# Improved Thermal and Electromagnetic Shielding of PEEK Composites by Hydroxylating PEK-C Grafted MWCNTs

**DOI:** 10.3390/polym14071328

**Published:** 2022-03-25

**Authors:** Fengyu Wen, Shu Li, Rui Chen, Yashu He, Lei Li, Lin Cheng, Jierun Ma, Jianxin Mu

**Affiliations:** Key Laboratory of High Performance Plastics, Ministry of Education, National & Local Joint Engineering Laboratory for Synthesis Technology of High Performance Polymer, College of Chemistry, Jilin University, 2699 Qianjin Street, Changchun 130012, China; wenfy19@mails.jlu.edu.cn (F.W.); lishu17@mails.jlu.edu.cn (S.L.); chenrui20@mails.jlu.edu.cn (R.C.); yshe20@mails.jlu.edu.cn (Y.H.); leili20@mails.jlu.edu.cn (L.L.); chenglin21@mails.jlu.edu.cn (L.C.); jrma21@mails.jlu.edu.cn (J.M.)

**Keywords:** thermal and electrical conductivity, electromagnetic interference shielding, surface treatments, composites

## Abstract

With the rapid rise of new technologies such as 5G and artificial intelligence, electronic products are becoming smaller and higher power, and there is an increasing demand for electromagnetic interference shielding and thermal conductivity of electronic devices. In this work, hydroxyphenolphthalein type polyetherketone grafted carboxy carbon nanotube (PEK-C-OH-g-MWCNTs-COOH) composites were prepared by esterification reaction. The composites exhibited good thermal conductivity, and compared with (MWCNTs-COOH/PEEK) with randomly distributed fillers, (PEK-C-OH-g-MWCNTs-COOH) composites showed a significant advantage, with the same carbon nanotube content, the thermal conductivity of PEK-C-OH-g-MWCNTs-COOH/PEEK (30 wt%) was 0. 71 W/(m-K), which was 206% higher than that of PEEK and 0.52 W/(m-K) higher than that of MWCNTs-COOH/PEEK (26.1 wt%). In addition, the PEK-C-OH-g-MWCNTs-COOH) composite exhibited excellent electrical conductivity and electromagnetic shielding (SE). The SE of 30 wt% PEK-C-OH-g-MWCNTs-COOH/PEEK is higher than the commercially used standard whose value is 22.9 dB (8.2 GHz). Thus, this work provides ideas for the development of thermally conductive functionalized composites.

## 1. Introduction

With the development of miniaturization and high frequency of electronic products, electromagnetic radiation and heat accumulation problems generated by electronic components during operation are becoming more and more serious, which not only deteriorating the performance of peripheral precision equipment, but also threatening human health. In order to effectively dissipate heat and minimize the damage caused by electromagnetic interference (EMI), materials with both good EMI shielding and thermal conductivity are urgently needed [1,2,3,4]. Compared with traditional metal-based EMI shielding heat dissipation materials, polymeric materials are of increasing interest for their light weight, flexibility, easy processing, and excellent corrosion resistance. However, most polymers inherently are poor conductors of heat and electrons, resulting in low EMI shielding effectiveness (SE) and thermal conductivity (TC) in polymeric matrices [5,6,7]. How to significantly improve the EMI SE and TC of polymer matrix at the same time is a key issue to promote the miniaturization and high frequency development of electronic products. Incorporating fillers with high electrical conductivity and high TC in polymer matrix is considered as one of the most feasible and effective ways to simultaneously improve the EMI SE and TC of polymer matrix. In particular, carbon nanomaterials such as graphene and carbon nanotubes (CNTs) have been widely investigated in polymer EMI shielding and heat dissipation composites due to their excellent conductivity and thermal conductivity [8,9].

In general, the carbon fillers are incompatible with the polymer matrix, resulting in serious aggregation of carbon filler in polymer composites and high thermal resistance at the filler-matrix interface [10,11,12], which greatly restricts the electromagnetic shielding, electromagnetic interference and thermal resistance properties of the prepared composites. However, due to the chemically inert surface of CNTs, coupled with the strong van der Waals forces inside, carbon nanotubes are interdigitated in the matrix or solution. The intertwined agglomerates in the matrix or solution are not easily dispersed, which substantially limits the thermal conductivity of CNTs. Therefore, enhancing the dispersion of CNTs in the matrix by modifying them functionally has been the main idea in the recent studies of CNTs. Lloyd’s research team [13] from the University of Notre Dame, USA, by using graphene(alkene)/polymers as an example investigated the influencing factors of interfacial thermal resistance theoretically and systematically using molecular dynamics simulations. The results show that the filler size, interfacial bonding strength and polymer density are the key factors for interfacial heat conduction, especially the interfacial bonding strength. When other conditions are consistent, the stronger the interaction between graphite (alkene) and matrix, the higher the thermal conductivity of the composite, and when the two are covalently bonded between graphite (alkene) and matrix, the best thermal conductivity of the composite is obtained at this time. Ding et al. [14] converted carbon nanotubes into graphene nanoribbons (GNR), and then performed a series of reactions with caprolactam and amino-caproic acid to obtain polyamide (PA)/GNR, and the results showed that the thermal conductivity of PA/GNR was improved by 165% and 95% in the in-plane and out-of-plane directions, respectively, at a GNR filling of only 0.5 wt% compared to pure PA. The covalently grafted GNR significantly reduced the number of thermal contacts between GNR layers and reduced the interfacial thermal resistance. Zhang et al. [15] grafted graphene oxide as well as hydroxylated h-BN onto PEG and increased the thermal conductivity by 71% compared to pure PEG, reaching to 0.574 W∙m^−1^∙K^−1^.

However, common polymer materials such as PA and PMMA can withstand low maximum temperature and poor chemical corrosion resistance, and cannot be used in some extreme cases, which greatly limits the use of prepared materials in more high-end complex fields. used, such as in aerospace. In order to prepare materials that can be used at higher temperatures and have better chemical corrosion resistance, we chose PEEK as the matrix. Due to its excellent chemical corrosion resistance and good thermal stability, PEEK material is widely used in the fields of automobile and aerospace, and is a promising thermoplastic composite material matrix [16]. Continuous service temperature up to 260 °C, much higher than PA, PMMA, etc. However, due to the insoluble nature of polyether ether ketone, it is difficult to directly graft chemical bonds with carbon materials. This leads to two possible phase separation phenomena, which are incompatibility between carbon nanotubes and PEEK, leading to agglomeration of carbon nanotubes and affecting the formation of thermally conductive network. In this work, MWCNTs were covalently grafted with hydroxyphenolphthalein-type polyether ketones similar to PEEK molecular structural unit to improve the dispersion of MWCNTs in PEEK and improve the interface with PEEK, thereby reducing the interfacial thermal resistance and improving the thermal conductivity of MWCNTs/PEEK composites. We firstly reduced phenolphthalein polyetherketone to obtain hydroxyphenolphthalein polyetherketone, then grafted hydroxyphenolphthalein polyetherketone to MWCNTs-COOH surface to prepare hydroxyphenolphthalein polyetherketone grafted carbon nanotubes, and then compounded the grafted carbon nanotubes with PEEK to prepare hydroxyphenolphthalein polyetherketone grafted carbon nanotube/PEEK composites, and investigated in detail the effects of carbon nanotubes before and after modification on the structure, thermal stability, electrical conductivity, electromagnetic shielding efficiency and thermal conductivity of polyetheretherketone-based composites.

## 2. Materials and Methods

### 2.1. Materials

PEEK powder (particle size:38 μm) with the melt index of 22 g/10 min was obtained from Changchun Jilin University Super Engineering Plastics Research Co. Ltd., Changchun, China. Multi-walled carbon nanotubes (MWCNTs, length: 10–30μm, external diameter: 5–15 nm, Internal diameter:2–5 nm,) was provided by Chinese Academy of Sciences Chengdu Organic Chemicals Co., Ltd. (Chengdu, China) and Phenolphthalein polyether ketone (PEK-C, PAEKNM-01) was purchased from Zhejiang Palco New Material Co., Ltd. (Zhengjiang, China). Sodium borohydride (NaBH_4_, AR), 4-Dimethylaminopyridine (DMAP, AR) and Ethanol(C_2_H_5_OH, AR) were supplied by Sinopharm Chemical Reagent Co., Ltd. (Tianjin, China). N,N-Dimethylformamide (DMF, AR), Dimethyl sulfoxide(DMSO, AR) and Isopropyl Alcoho (C_3_H_8_O, ACS)were obtained from Shanghai Aladdin Bio-Chem Technology Co., Ltd. (Shanghai, China). N,N′-Dicyclohexylcarbodiimide (DCC, AR) and Hydrochloric acid(HCl, AR) were provided by Shanghai Maclean Bio-chemical Technology Co., Ltd. (Shanghai, China). All the above reagents can be used without further purification.

### 2.2. Method

X-ray photoelectron spectroscopy (XPS, THERMO FISHER SCIENTIFIC ESCALAB 250, America) was employed to examine the composition of the modified fillers. Crystal integrity of carbonaceous materials were inspected via Raman spectra (LabRAM, HR800, America). Scanning electron microscopy (SEM, HITACHI SU8020, America), transmission electron microscope (TEM, TECNAI F20, Japan), were utilized for observing surface appearance and the profile of composites. The infrared spectra were collected by using a Fourier transform infrared spectrometer (Nicolet 5PC, America). Samples were prepared by mixing KBr dry powder. FTIR spectra were obtained between 4000 and 500 cm^−1^. The thermal stability analysis of composites was carried out using a thermogravimetric analyzer (Pryis 1 TGA, Perkin Elmer, America) under nitrogen atmosphere. A differential scanning calorimeter (DSC Q2000, Thermo-VG Scientific, America) was used from 100 to 400 °C at a rate of 10 °C/min under a nitrogen atmosphere. NMR spectrum were obtained using a spectrometer (Bruker-Avance III spectrometer 400 MHz, BRUKER, Germany). Electromagnetic shielding performance of composites with 11 mm × 2.4 mm × 2.5 mm dimensions was measured by microwave network analyzer (Agilent N5244A PNA-X, America), and frequency range of electromagnetic was 8.2–12.4 GHz wave. Laser flash thermal conductivity meter (LFA 467, Germany) was used to test the thermal conductivity of composite materials.

### 2.3. Composites Synthesis

#### 2.3.1. Synthesis of Hydroxyphenolphthalein Polyether Ketones

The hydroxylation of PEK-C material was achieved by reducing the stable carbonyl group in the PEK-C molecule to the more reactive hydroxyl group using a strong reducing agent, NaBH4 [17], as shown in Figure 1a. The specific steps are as follows: 3 g NaBH4 was added to anhydrous 250 mL DMSO solution and stirred under nitrogen atmosphere with a magnetic stirrer until sufficient NaBH4 was dissolved. Then add 25 g PEK-C powder into the above solution and react under 120 °C and nitrogen atmosphere for 1 h. After stopping the heating, the material was cooled naturally, discharged to isopropanol and washed with ethanol, dilute hydrochloric acid and distilled water repeatedly for 3 times in turn to complete the preparation of hydroxylated PEK-C material, which was recorded as PEK-C-OH.

#### 2.3.2. Synthesis of Hydroxyphenolphthalein Polyetherketone Grafted Carbon Nanotubes

1.0 g MWCNTs-COOH was added to 200 mL anhydrous DMF solution and ultrasonically dispersed, then 3.0 g PEK-C-OH, 0.45 g DMAP and 3.0 g DCC were added to 100 mL anhydrous DMF solution, and after full dissolution, DMF solution of MWCNTs-COOH was added to it, and the reaction was carried out under nitrogen atmosphere at 50 °C for 72 h. The reaction solution was filtered, and the product was washed with DMF several times to wash off the excess DMAP, DCC and the PEK-C-OH. Finally, the PEK-C-OH grafted carboxylated carbon nanotubes (PEK-C-OH-g-MWCNTs-COOH) were synthesized by drying in a vacuum oven at 70 °C for 24 h. The preparation process is shown in Figure 1b.

#### 2.3.3. Synthesis of Hydroxyphenolphthalein-Based Polyetheretherketone Grafted Carbon Nanotube/Polyetheretheretherketone Composites

MWCNTs-COOH-g-PEK-C-OH was added to ethanol for ultrasonic dispersion for 2 h. After that, PEEK powder was added for mechanical stirring for 12 h. The ethanol volume to PEEK powder mass ratio was 0.04 L/g. The above suspension was subsequently filtered and dried at 120 °C for 3 h to obtain MWCNTs-COOH-g-PEK-C-OH/PEEK premix, which was subsequently made into 0.6 mm thick composite samples using a press at 370 °C and 20 MPa pressure. The preparation process is shown in Figure 1c.

## 3. Results and Discussion

### 3.1. Characteristics of Specimens

According to the literature, it is known that the chemical shift of H on the benzene ring is generally concentrated between 6 to 8 ppm, while PEK-C-OH after hydroxylation at 5.7 ppm has one more peak than PEK-C before hydroxylation as observed from Figure 2a, which is the chemical shift peak of H in the diphenylmethylene group, and since in PEK-C-OH there is a joint influence of the benzene ring and the hydroxyl group, this group of peaks moves to the lower field. It further confirms that the carbonyl group is reduced to hydroxyl group.

As observed from the Figure 2b, the absorption peak of the carbonyl group of aromatic ketone (Ar-CO-Ar) is observed at 1650 cm^−1^ in PEK-C, while in the FT-IR spectrum of PEK-C-OH, the peak at 1650 cm^−1^ has largely disappeared, along with the characteristic absorption peaks of C=O [18,19] of the five-membered lactone rings in PEK-C and PEK-C-OH observed at 1766 cm^−1^. The wave peak at 3500 cm^−1^ in PEK-C-OH became larger, which indicated that PEK-C did produce a large amount of hydroxyl groups after reduction by NaBH4 and did not destroy the other molecular structures in PEK-C, which tentatively confirmed that we successfully prepared PEK-C-OH.

The FT-IR spectrum of MWCNTs-COOH and PEK-C-OH-g-MWCNTs-COOH can be seen in Figure 2c. The absorption peaks at 1694 cm^−1^ and 1625 cm^−1^ correspond to the C=O and C-O-C stretching vibrations, respectively, which are typical characteristic peaks of MWCNTs-COOH [20,21]. In compared to MWCNTs-COOH, PEK-C-OH-g-MWCNTs-COOH exhibits the characteristic absorption peaks of PEK-C-OH, the peaks at 2925 cm^−1^ and 2851 cm^−1^ correspond to C-H vibrations in PEK-C-OH. The characteristic peaks of stretching vibrations on the benzene ring of PEK-C-OH are noted at 1576 cm^−1^ and 1535 cm^−1^. The asymmetric stretching vibration peak of the Ar-O-Ar bond in PEK-C-OH are observed at 1244 cm^−1^, along with the characteristic absorption peak of C=O in the five-membered lactone ring of PEK-C-OH can be observed at 1764 cm^−1^. From the FT-IR spectrum of PEK-C-OH -g-MWCNTs-COOH, the peak at 1725 cm^−1^ attributed to the carbonyl group (C=O) in the generated ester group, which is in agreement with previous reports in the literature, thus it can be demonstrated that the successful grafting of the PEK-C-OH molecular chain on MWCNTs-COOH.

It can be seen from Figure 2d, that the relative intensity of the Raman curve of PEK-C-OH-g-MWCNTs-COOH is higher than that of MWCNTs-COOH, which may be due to the destruction of the MWCNTs-COOH structure after grafting. In Raman spectrum, the ratio of the D-peak intensity (ID) to the G-peak intensity (IG), ID/IG, can indicate the degree of defects of the carbon material, and the ID/IG value of PEK-C-OH-g-MWCNTs-COOH after grafting is higher than that of MWCNTs-COOH, as shown in Table 1. This is because PEK-C-OH works on the electronic properties of MWCNT-COOH, which in turn may have led to an increase in ID/IG [22,23], confirming the success of modifying MWCNTs-COOH.

To further verify that PEK-C-OH was grafted on the surface of carboxyl carbon nanotubes, we performed XPS analysis on two carbon nanotubes, MWCNTs-COOH and PEK-C-OH-g-MWCNTs-COOH, as shown in Figure 3a.The curves were fitted with a Gaussian function, and the C1s spectrum of MWCNTs-COOH and PEK-C-OH-g-MWCNTs-COOH were shown in Figure 3b,c The C1s spectrum of MWCNTs-COOH can be divided into five peaks with corresponding binding energies of 284.6 eV, 285.6 eV, 286.8 eV, 289.0 eV, and 291.5 eV. where the binding energy 284.6 eV corresponds to the sp2 structure of C in the graphite structure (C-C) [24], the peak at 285.6 eV corresponds to the sp3 structure of C (a small number of structural defects formed in the preparation of carbon nanotube) [25], the peak at 286.8 eV corresponds to the binding energy of 1s electrons when C is bound to one O atom (C-OH) [26,27], the peak at 289.0 eV corresponds to the binding energy of the 1 s electron when C is bound to two O atoms (-COOH) [28], and 291.5 eV was the satellite peak generated by the π-π* electronic transition, when the high binding energy of the main peak is at 284.5 eV [29,30]. As shown in Table 2 and Table 3, the ratio of nO/nC of the modified PEK-C-OH-g-MWCNTs-COOH increased, the proportion of -COOH decreased, and the percentage of C-OH increased, which further proved that PEK-C-OH was successfully grafted to the MWCNTs-COOH surface.

The thermogravimetric analysis of MWCNTs-COOH and PEK-C-OH-g-MWCNTs-COOH were performed in the nitrogen atmosphere. As observed from the Figure 4a, MWCNTs-COOH was thermally stable, and the mass loss was only 2.7% in the temperature range of 100–800 °C, which can be regarded as the content of carboxyl functional groups. While PEK-C-OH-g-MWCNTs-COOH showed significant thermal mass loss in the above temperature range with the mass loss of about 13%, which can be regarded as the content of PEK-C-OH. Combined with the DTG curve analysis in Figure 4b, it was found that the temperature at which PEK-C-OH-g-MWCNTs-COOH showed significant mass loss was higher, and the peak appeared at around 493 °C, which was mainly due to the thermal decomposition of PEK-C-OH grafted on the surface of MWCNTs-COOH, The peak appeared at approximately the same location as the thermal decomposition weight loss peak of PEK-C-OH (at around 497 °C), which again confirmed the success of the MWCNTs-COOH surface grafting. As can be seen from Figure 4a, PEK-C exhibited excellent thermal stability with the onset thermal decomposition temperature (defined as the temperature corresponding to 5% thermal mass loss) as high as 488 °C, while PEK-C-OH had the onset thermal decomposition temperature of only 328 °C due to the presence of -OH. PEK-C-OH-g-MWCNTs-COOH exhibited relatively high thermal stability, and its initial thermal decomposition temperature was 471 °C, which fully meets the processing temperature of PEEK (about 370 °C). When preparing 5 wt%, 10 wt%, 20 wt% and 30 wt% PEK-C-OH-g-MWCNTs-COOH/PEEK composites in turn, considering the grafting rate of 13%, the actual content ratio of carbon nanotubes should correspond to 4.35 wt%, 8.7 wt%, 17.4 wt% and 26.1 wt% in sequence, so that in the preparation of MWCNTs-COOH/PEEK composites, the carbon nanotubes should be dosed according to the above carbon nanotube ratios.

It can be seen from Figure 5 that the tube length of MWCNTs-COOH was relatively long and the surface was relatively smooth, but the entanglement between tubes was particularly serious. In contrast, the distribution between tubes in PEK-C-OH-g-MWCNTs-COOH was more dispersed and the entanglement phenomenon was greatly reduced. To further observe the surface roughness of both carbon nanotubes before and after grafting, TEM tests were performed, and it can be seen from Figure 6 that the diameter of carbon nanotubes increased after modification, and the surface roughness also increased, which further confirmed that PEK-C-OH was successfully grafted to MWCNTs-COOH. However, it can also be seen from the figure that the PEK-C-OH grafting was not homogeneous. This was caused by the poor dispersion of carbon nanotubes in the system during the reaction, which cannot fully react with the reactant PEK-C-OH.

### 3.2. Composite Performance

Figure 7 was the SEM images of MWCNTs/PEEK composites. Compared with the MWCNTs-COOH/PEEK composites, it can be observed that the interface between PEK-C-OH-g-MWCNTs-COOH and the PEEK matrix was blurred and dispersed more homogeneous, and no obvious aggregation phenomenon of PEK-C-OH-g-MWCNTs-COOH was found, indicating that the compatibility of the modified carbon nanotubes with the PEEK matrix was increased. Before grafting, the unmodified carbon nanotubes may agglomerate due to the van der Waals force between the carbon nanotubes and the phase separation effect caused by the low compatibility of the two, while the grafted carbon nanotubes and PEEK The compatibility is improved and the occurrence of agglomeration is reduced [30,31,32]. By chemical grafting, the aggregation of MWCNTs-COOH was significantly weakened, so that MWCNTs-COOH can be relatively uniformly dispersed in the PEEK matrix, which facilitates sufficient lapping between the fillers and the formation of a large number of thermally conductive and electron-conductive networks, which is conducive to the transport of phonons and electrons.

#### 3.2.1. Thermal Performance Analysis of MWCNTs/PEEK Composites

As shown in Figure 8a,b, the onset thermal decomposition temperature (the temperature corresponding to 5% mass loss) and the temperature of 10% mass loss of pure PEEK were 558 °C and 567 °C, respectively, while the onset thermal decomposition temperature and the temperature of 10% mass loss of the 5 wt%modified PEK-C-OH-g-MWCNTs-COOH/PEEK composites were 562 °C and 569 °C, respectively. Compared with pure PEEK, it increased by 4 °C and 2 °C, respectively. When the mass fraction of MWCNTs-COOH was 4.35%, the onset thermal decomposition temperature and the 10% mass loss of of MWCNTs-COOH/PEEK were 551 °C and 559 °C, respectively, which decreased by 7 °C and 8 °C, respectively, compared with pure PEEK. The thermal stability of MWCNTs-COOH/PEEK was decreased and the thermal decomposition rate was faster than that of pure PEEK, indicateing that the addition of too much carbon nanotubes did not improve the thermal stability performance of the composites, which was due to the high content of carbon nanotubes leading to difficulties of filler dispersion, resulting in changes in the structure and properties of the composites. This also indicates that the modified PEK-C-OH-g-MWCNTs-COOH/PEEK can effectively improve the thermal stability of their composites due to the better dispersion and interfacial bonding of the modified PEK-C-OH-g-MWCNTs-COOH/PEEK in the PEEK matrix [33].

The DSC curve of MWCNTs/PEEK composites are shown in Figure 8c. From the DSC curve and Table 4, it can be seen that the addition of MWCNTs-COOH resulted in a decrease in the melting point and a small increase in the glass transition temperature of the composites, which was made more pronounced by PEK-C-OH-g-MWCNTs-COOH. -COOH in the MWCNTs-COOH molecules had weak intermolecular interactions with the PEEK molecular chains, which reduces the intermolecular entanglement of PEEK molecules and improves the mobility of PEEK molecular chains. In contrast, PEK-C-OH in the PEK-C-OH-g-MWCNTs-COOH molecule further reduces the interfacial energy and promotes intermolecular interactions between PEEK molecules, which ultimately leads to a lower melting point of the composite. However, the interfacial enhancement enhances PEK-C-OH-g-MWCNTs-COOH to impede the motion of PEEK molecular chain segments, which increases the glass transition temperature of the composites.

#### 3.2.2. Thermal Conductivity Analysis of MWCNTs/PEEK Composites

The thermal conductivity of MWCNTs/PEEK is represented in Figure 9a. The thermal conductivity of MWCNTs/PEEK increased with the addition of MWCNTs-COOH, while the thermal conductivity of the prepared PEK-C-OH-g-MWCNTs-COOH/PEEK was better with the addition of MWCNTs-COOH. When the PEK-C-OH-g-MWCNTs-COOH content was 20 wt% and 30 wt%, respectively, the out-of-plane thermal conductivity of the composites was 0.64 W/(m-K) and 0.71 W/(m-K), respectively, which was 176% and 206% higher than that of pure PEEK, while the The in-plane thermal conductivity of 30 wt% PEK-C-OH-g-MWCNTs-COOH/PEEK composites was up to 1.06 W/(m-K). In contrast, the thermal conductivity of the composites was only 0.47 W/(m-K) and 0.52 W/(m-K) for MWCNTs-COOH content of 17.4 wt% and 26.1 wt%, respectively, which also means that at both carbon nanotube content of 26.1 wt%, the modified carbon nanotubes/PEEK thermal conductivity was improved by 36.5% over the unmodified carbon nanotube/PEEK composite. The reasons are as follows, firstly, due to the grafting of PEK-C-OH-g-MWCNTs-COOH on the surface, which improved the dispersion of MWCNTs-COOH in the PEEK matrix, the well-dispersed PEK-C-OH-g-MWCNTs-COOH was able to form a more effective thermal conductivity network, and secondly, the PEK-C-OH-g-MWCNTs-COOH surface grafted PEK-C-OH grafted on the surface of MWCNTs-COOH improved the interface between MWCNTs-COOH and PEEK substrate and increased the interfacial bonding between MWCNTs-COOH and PEEK, thus reducing the interfacial thermal resistance between MWCNTs-COOH and PEEK substrate and decreasing the scattering probability of interfacial phonons, as shown in Figure 9b. The increase in PEK-C-OH-g-MWCNTs-COOH/PEEK thermal conductivity when the PEK-C-OH-g-MWCNTs-COOH content is increased from 0 to 10 wt% is probably due to the random distribution of PEK-C-OH-g-MWCNTs-COOH in PEEK when the filler content is less than 10 wt%. The contact probability between carbon nanotubes is very low, and when it increases to 10 wt%, a large number of contact points appear between carbon nanotubes and more thermal conductivity networks are constructed, after which the filler then increases the PEK-C-OH-g-MWCNTs-COOH content, and although a more complex thermal conductivity network is formed, more contact points also mean an increase in the thermal resistance of the contact between the fillers, making the rate of increase in thermal conductivity is not significant. A similar situation exists for MWCNTs-COOH/PEEK.

#### 3.2.3. Electrical Conductivity and Electromagnetic Shielding Effectiveness Analysis of MWCNTs/PEEK Composites

MWCNTs possess good electrical conductivity, and we tested the conductivity of the composites. As expected, the conductivity of the composites increased significantly with the addition of MWCNTs, as shown in Figure 10. The conductivity of the pure PEEK sheet (insulator), which has a very low conductivity, increased from 3.1 × 10^−6^ S/m (5 wt%) to 3.0 S/m (30 wt%) with the addition of PEK-C-OH-g-MWCNTs-COOH to the composite. In contrast, the conductivity of the composites with MWCNTs-COOH addition was relatively low, and the conductivity of their composites increased from 2.0 × 10^−6^ S/m (5 wt%) to 1.3 S/m (26.1 wt%). This is due to the grafting of PEK-C-OH on the surface of PEK-C-OH-g-MWCNTs-COOH, which improved the dispersion of MWCNTs-COOH in the PEEK matrix, and the well-dispersed PEK-C-OH-g-MWCNTs-COOH was able to form an effective conductive network. Similarly, the strong interfacial bonding between the modified MWCNTs-COOH and PEEK can reduce the tunnel resistance between adjacent MWCNTs-COOH, thus improving the electrical conductivity of MWCNTs/PEEK. The electrical conductivity of the composites is of percolation type. For its mathematical description, the following equation of the classical theory of percolation was adopted [34]
(1)$$\sigma =\sigma_{0}(\varphi -\varphi_{c}{)}^{t}$$
where *σ*_0_ and *σ* are the conductivity of the filler and composite, respectively (S/cm), *φ* and *φ_c_* are the volume fractions of the filler and the filler at the percolation threshold, and t is the critical indicator of the electrical conductivity (dimensionless constant). The electric percolation values obtained by formula transformation and fitting are 3.36 vol.% (5 wt%) and 2.92 vol.% (4.5 wt%), as shown in Figure 10.

The electromagnetic shielding effectiveness of MWCNTs/PEEK composite (thickness = 0.6 mm) at X-band is shown in Figure 11a. The electromagnetic shielding effectiveness of pure PEEK is almost 0 dB, indicating that it has no reflection and absorption of electromagnetic waves. The total electromagnetic shielding effectiveness value of the PEEK-based composite increases with the increase of filler content and is relatively constant throughout the frequency range with little fluctuation, indicating that the material has a wide absorption bandwidth for electromagnetic waves. Due to the high conductivity of carbon nanotubes, the electromagnetic shielding effectiveness of PEK-C-OH-g-MWCNTs-COOH/PEEK increased from 0.13 dB (0 wt%) to 22.9 dB (30 wt%) at 8.2 GHz with the increase of PEK-C-OH-g-MWCNTs-COOH content, as shown in Figure 11b, while the electromagnetic shielding effectiveness of MWCNTs-COOH/PEEK with MWCNTs-COOH as filler has a relatively low electromagnetic shielding effectiveness, which only increases from 0.13 dB (0 wt%) to 17.4 dB (26.1 wt%). At 8.2 GHz, the contributions of absorption loss (SEA) and reflection loss (SER) to the total electromagnetic shielding effectiveness value are shown in Figure 11c,d. The SEA and SER values increase with the increase of carbon nanotube content, and the SER contribution value is higher at low filler content, but the SEA contribution value of the composite is higher with the increase of filler content, realizing that the contribution is dominated by SER dominant to SEA dominant transition, and the PEK-C-OH-g-MWCNTs-COOH/PEEK composite can achieve this transition more quickly, which may be due to the better dispersion of CNTs in favor of the composite absorption loss increases. Table 5 shows the electromagnetic shielding properties of various composite materials. The electromagnetic shielding properties of the composite materials prepared by us are better than these materials.

The improved electrical conductivity and electromagnetic shielding capabilities demonstrate the completion of the transition from insulator to semiconductor in polyetheretheretherketone composites and the formation of a substantial conductive network in the matrix, promising a potential alternative for antistatic and EMI shielding applications.

## 4. Conclusions

In this work, PEK-C-OH-g-MWCNTs-COOH was prepared by grafting with MWCNTs-COOH after reduction of PEK-C. The interfacial properties were improved by the chemical bonding between PEK-C-OH and MWCNTs-COOH and good compatibility between PEK-C-OH-g-MWCNTs-COOH and PEEK. The SEM characterization showed that PEK-C-OH-g-MWCNTs-COOH was more dispersed in the composite than MWCNTs-COOH before modification. After the PEK-C with high compatibility with PEEK is grafted with carbon nanotubes, the compatibility between carbon tubes and polyetheretherketone is improved, the effect of phase separation is reduced, and the chemical grafting. The interfacial thermal resistance between the carbon nanotubes and the polymer is reduced, which effectively improves the thermal conductivity of the composite material. 30 wt% PEK-C-OH-g-MWCNTs-COOH/PEEK was prepared with a thermal conductivity of 0.71 W/(m-K), which was 206% higher than that of the thermal conductivity of 30 wt% PEK-C-OH-g-MWCNTs-COOH/PEEK was 0.71 W/(m-K), an improvement of 206% compared to PEEK, while the thermal conductivity of 26.1 wt% MWCNTs-COOH/PEEK at the same carbon nanotube content was only 0.52 W/(m-K). Composites also improved in electrical conductivity and electromagnetic shielding ability from 26.1 wt% MWCNTs-COOH/PEEK 1.3 S/m, 17.4 dB to 30 wt% PEK-C-OH-g-MWCNTs-COOH/PEEK 3.0 S/m, 22.9 dB. The excellent properties obtained from PEEK composites prepared by this modification method are urgently needed for current microelectronic applications.

## Figures and Tables

**Figure 1 polymers-14-01328-f001:**
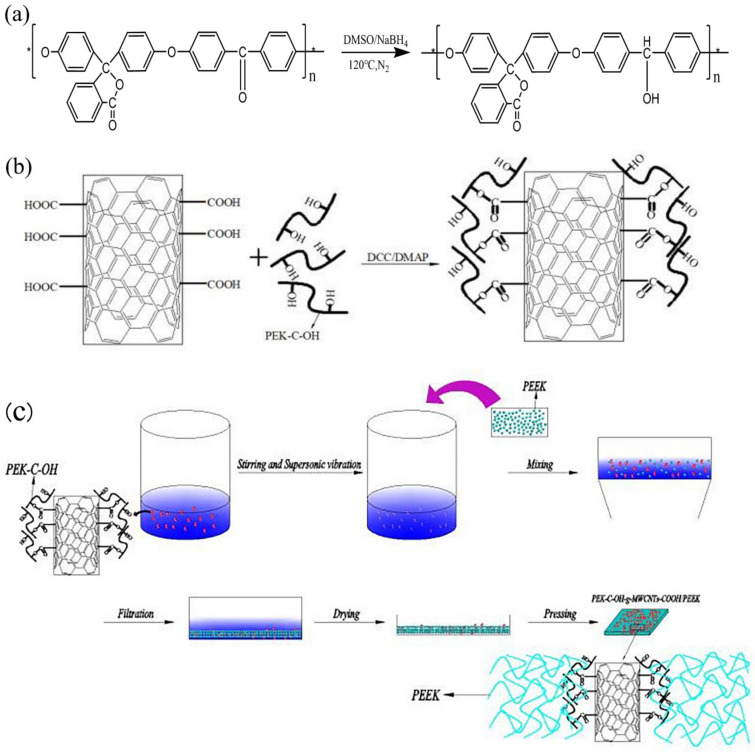
(**a**) Synthesis of PEK-C-OH; (**b**) Synthesis of PEK-C-OH-g-MWCNTs-COOH; (**c**) The preparation process of MWCNTs-COOH-g-PEK-C-OH/PEEK composites.

**Figure 2 polymers-14-01328-f002:**
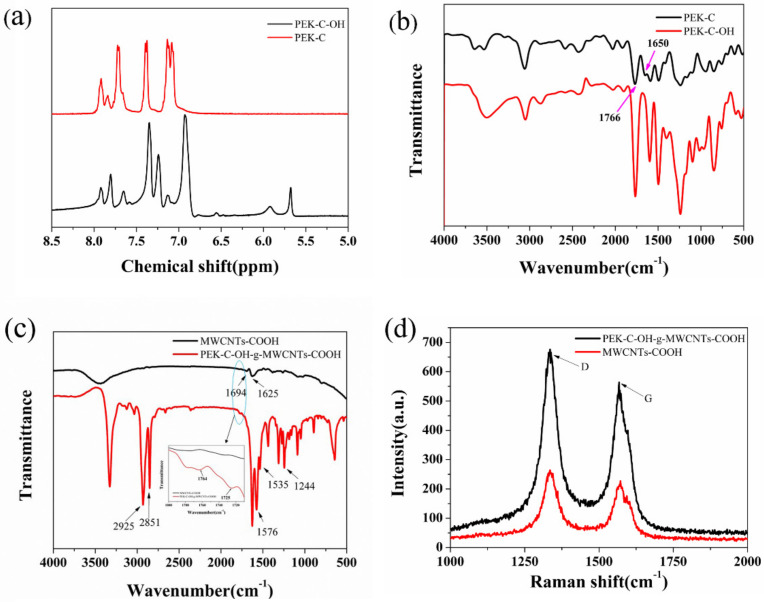
(**a**) ^1^H NMR spectra of PEK-C and PEK-C-OH; (**b**) Infrared spectra of PEK-C and PEK-C-OH; (**c**) Infrared spectra of MWCNTs-COOH and PEK-C-OH-g-MWCNTs-COOH; (**d**) Raman spectra of MWCNTs-COOH and PEK-C-OH-g-MWCNTs-COOH.

**Figure 3 polymers-14-01328-f003:**
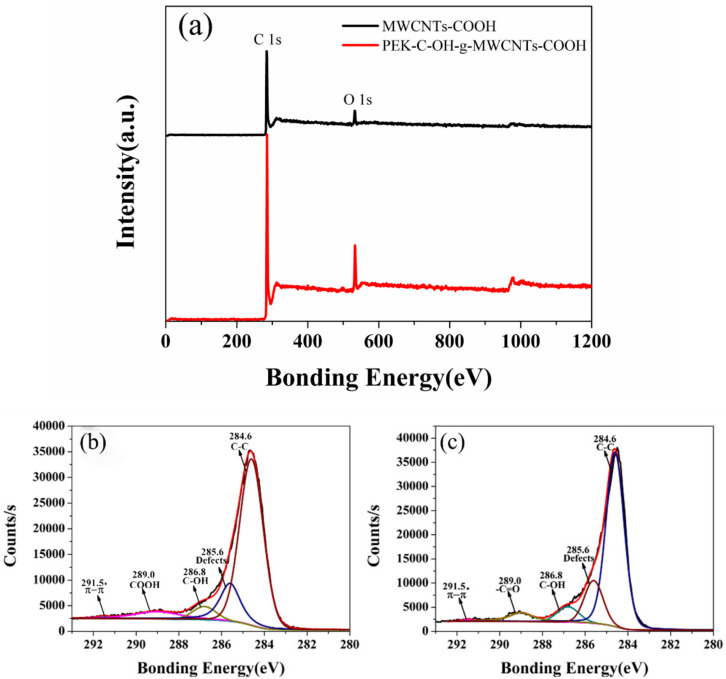
(**a**) XPS wide scan spectrum of MWCNTs-COOH and PEK-C-OH-g-MWCNTs-COOH; (**b**) XPS C1s fine scan spectra of MWCNTs-COOH; (**c**) XPS C1s fine scan spectra of PEK-C-OH -g-MWCNTs-COOH.

**Figure 4 polymers-14-01328-f004:**
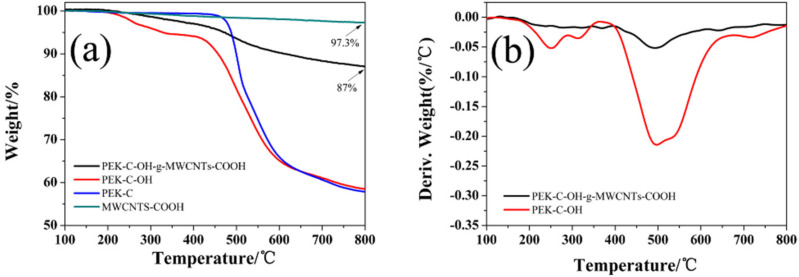
(**a**) TGA of PEK-C, PEK-C-OH, MWCNTs-COOH and PEK-C-OH-g-MWCNTs-COOH; (**b**) DTG of PEK-C-OH and PEK-C-OH-g-MWCNTs-COOH.

**Figure 5 polymers-14-01328-f005:**
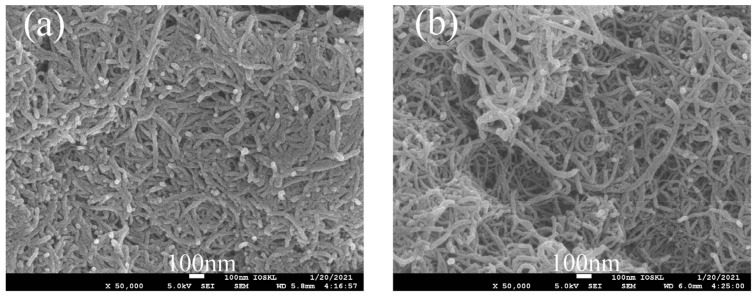
SEM images of (**a**) MWCNTs-COOH; (**b**) PEK-C-OH-g-MWCNTs-COOH.

**Figure 6 polymers-14-01328-f006:**
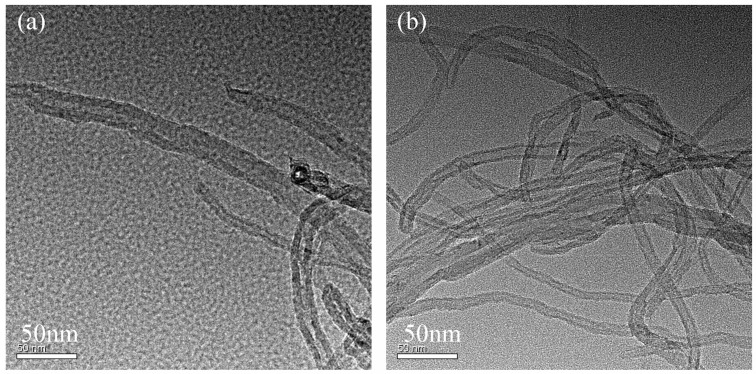
TEM images of (**a**) MWCNTs-COOH; (**b**) PEK-C-OH-g-MWCNTs-COOH.

**Figure 7 polymers-14-01328-f007:**
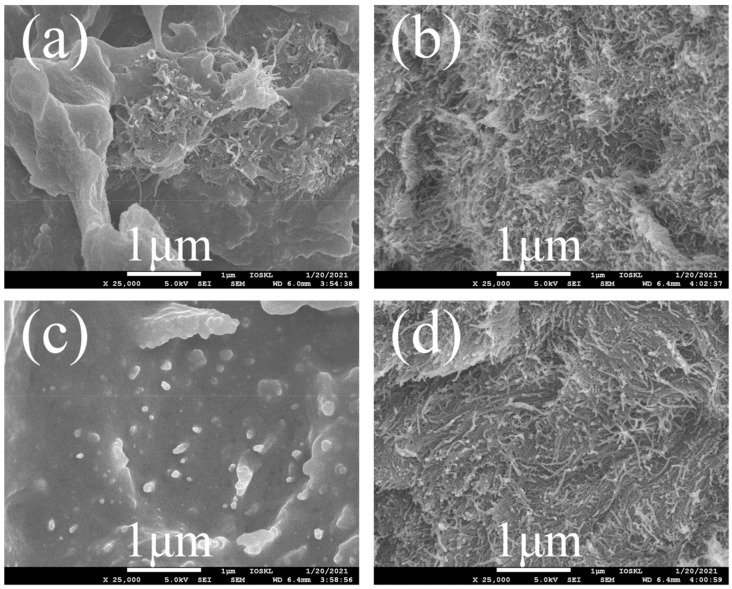
SEM images of (**a**) 10 wt% MWCNTs-COOH/PEEK; (**b**) 20 wt% MWCNTs-COOH/PEEK; (**c**) 8.7 wt% PEK-C-OH-g-MWCNTs-COOH/PEEK; (**d**) 17.4 wt% PEK-C-OH-g-MWCNTs-COOH/PEEK.

**Figure 8 polymers-14-01328-f008:**
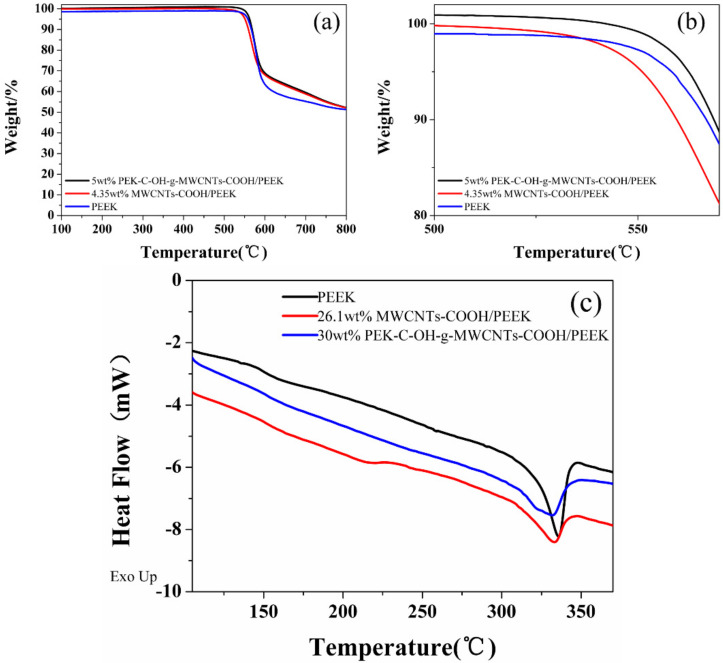
(**a**,**b**) TGA curvs of MWCNTs/PEEK composites; (**c**) DSC curvs of MWCNTs/PEEK composites.

**Figure 9 polymers-14-01328-f009:**
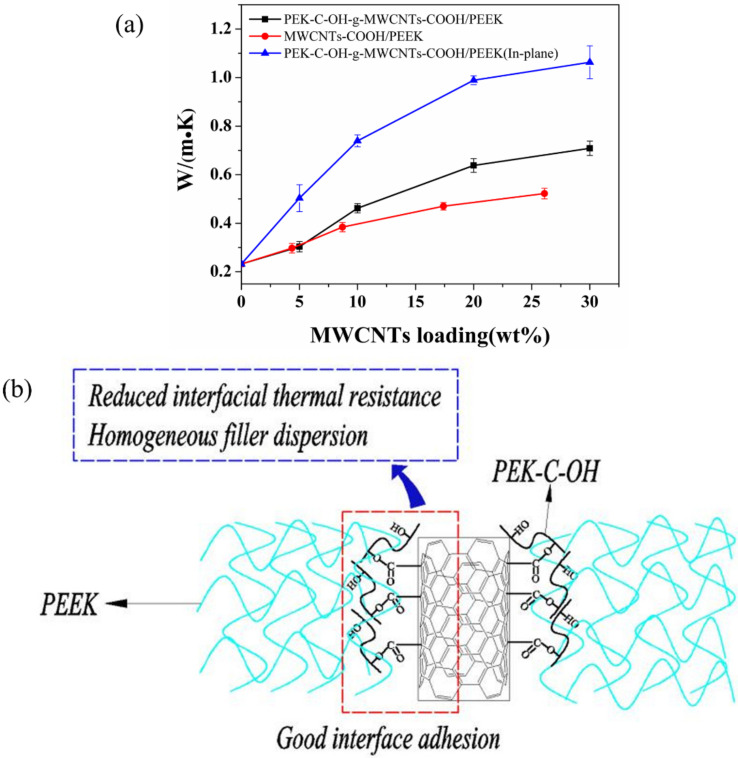
(**a**) Thermal conductivity of MWCNTs/PEEK composites; (**b**) The schematic diagram of the mechanism of increasing thermal conductivity of PEK-C-OH-g-MWCNTs-COOH/PEEK composites.

**Figure 10 polymers-14-01328-f010:**
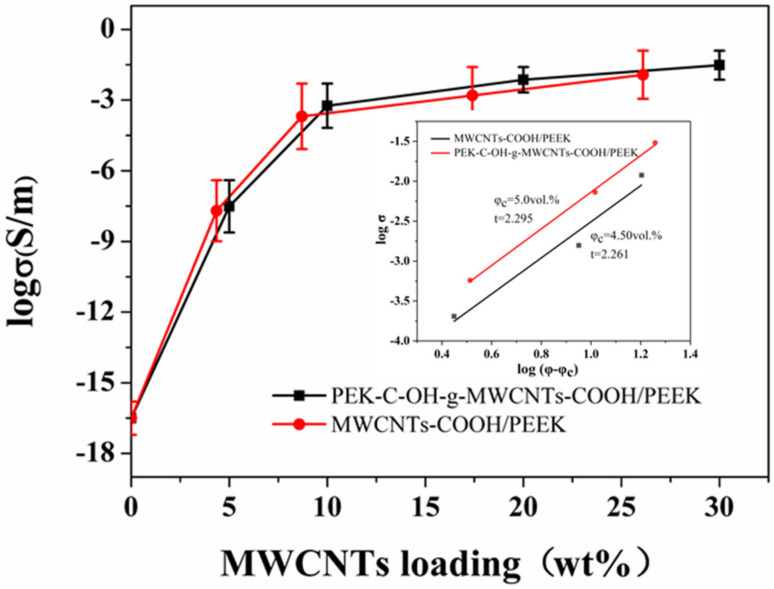
Electric conductivity of MWCNTs/PEEK composites.

**Figure 11 polymers-14-01328-f011:**
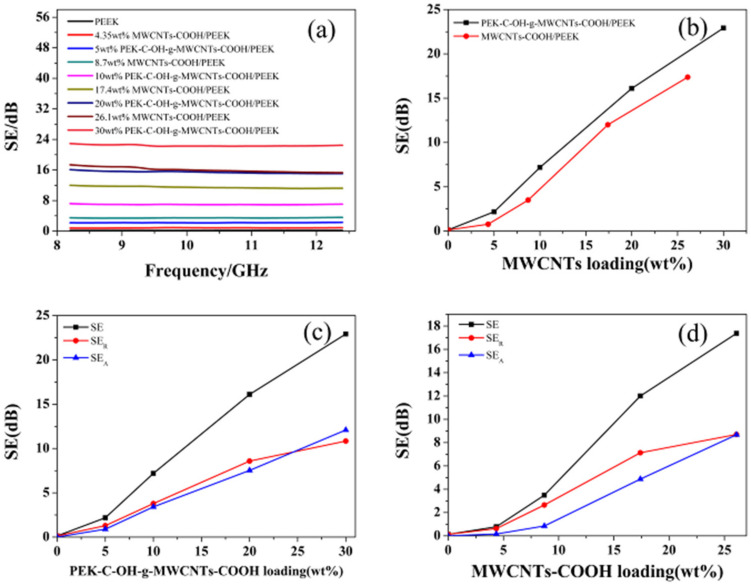
EMI SE of the MWCNTs/PEEK composites (thickness = 0.6 mm) in X-band: (**a**) total SE versus frequency; (**b**) total SE measured at 8.2 GHz versus logσ; (**c**) SE of PEK-C-OH-g-MWCNTs-COOH/PEEK composites at 8.2 GHz; (**d**) SE of MWCNTs-COOH/PEEK composites at 8.2 GHz.

**Table 1 polymers-14-01328-t001:** Raman spectra data of MWCNTs-COOH and PEK-C-OH-g-MWCNTs-COOH.

Samples	D/cm^−1^	G/cm^−1^	ID	IG	ID/IG
MWCNTs-COOH	1335.0	1572.2	24,808.3	23,174.2	1.07
PEK-C-OH-g-MWCNTs-COOH	1335.0	1567.5	61,982.3	55,392.0	1.12

**Table 2 polymers-14-01328-t002:** The surface element content of MWCNTs-COOH and PEK-C-OH-g-MWCNTs-COOH.

Samples	C1s(%)	O1s(%)	nO/nC
MWCNTs-COOH	91.96	8.04	0.087
PEK-C-OH-g-MWCNTs-COOH	88.59	11.41	0.12

**Table 3 polymers-14-01328-t003:** XPS results: C1s peak positions E, areas A and assignations.

MWCNTs-COOH		PEK-C-OH-g-MWCNTs-COOH		
E(eV)	A(%)	E(eV)	A(%)	Assignation
291.5	0.44	291.5	0.435	π-π
289.0	7.35	289	6.15	-COOH(-C=O)
286.8	5.27	286.8	6.88	C-OH
284.6	69.82	284.6	69.75	C-C
285.6	17.12	285.6	16.585	Defects

**Table 4 polymers-14-01328-t004:** Tg and Tm of MWCNTs/PEEK composites.

Samples	Tg/°C	Tm/°C
PEEK	148.63	335.66
26.1 wt%MWCNTs-COOH/PEEK	150.77	332.72
30 wt%PEK-C-OH-g-MWCNTs-COOH/PEEK	152.27	331.66

**Table 5 polymers-14-01328-t005:** Electromagnetic shielding effectiveness of various composite materials.

Shielding Material	EMI SE (dB)
MWCNT/cellulose [35]	20
MWCNT/polystyrene [36]	20
MWCNT/PANI [37]	7.1
Graphene/PMMA [38]	13–19
Graphene/Fe_3_O_4_/epoxy [39]	17
Graphene/Ag-mesh film/PMMA [40]	14.1

## Data Availability

The raw/processed data required to reproduce these findings cannot be shared at this time due to legal or ethical reasons.

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
