# Peer review of "Improved Thermal and Electromagnetic Shielding of PEEK Composites by Hydroxylating PEK-C Grafted MWCNTs"

_polymers, 2022, doi:10.3390/polym14071328_

Round 1

Reviewer 1 Report

The work «Improved thermal and electromagnetic shielding of PEEK composites by hydroxylating PEK-C grafted MWCNTs» is devoted very actual topic according modern development of miniaturization and high frequency of electronic products. Problem being solved and selected materials and methods to solve it are named in introduction. Synthesis method is illustrated using good quality figure. Complexes of modern informative research methods characterize samples produced. All figures are of good quality. The results obtained are of scientific and practical significance and prospects for further development. Key results are reflected in the conclusion. However, I have some questions:

  1. The authors state that the percolation threshold for PEK-C-OH-g-MWCNTs-COOH and MWCNTs-COOH in PEEK is about 5 wt%. This does not cause disputes, however, the authors should process the electrical conductivity data from the mass fraction of the filler according to the classical percolation law. This will make it possible to determine the differences in the percolation threshold for the two systems and compare them, as well as to make an assessment of the electrical conductivity of the filler and compare. You can use the method Composites Science and Technology, 108972 or any other.
  2. The discussion of the results in lines 383-403 must be supplemented by a comparison with similar systems - composites. Additionally, add information about the nature of the effect. Do all CNT composites exhibit electromagnetic shielding effectiveness?

Author Response

Modified content attachment sent to you

Reviewer 2 Report

Dear authors,

The article is very good. A very positive impression is made by the developed methodology and the analysis of the results - especially FTIR + for this correlation with TG. In principle, I have no comments on the methodology, results, analysis, discussion, and conclusions. In fact, I have only a few questions that I would like to ask you to answer:

1) why did you choose PEEK? it is very heavy material to process and in principle, it is better to take other structural ones i.e. PA, PMMA, I am curious why you chose PEEK?

2) intro is the only weakness of this article because:
- you did not write what is your motivation for this research, what is the problem, what do you relate your research to? I guess and it is quite understandable for me - but remember that those who read it can not always guess;

- in principle you do not write anything about the main "material" problem of this article - that is phase segregation of ceramic fillers in the polymeric matrix, the influence of surface modification of ceramics, the role of functional groups is very well addressed - but write at least 3 - 4 more sentences about phase segregation, the role of chemical bonds - not only the van der Waals forces; here you have an example article where you write about it:

(1) Molecular dynamics study of the penetration resistance of multilayer polymer/ceramic nanocomposites under supersonic projectile impacts

(2) Polymer-derived ceramic/graphene oxide architected composite with high electrical conductivity and enhanced thermal resistance

(3) Surface modification methods of ceramic filler in ceramic-carbon fibre composites for bioengineering applications - A systematic review

- Please make your conclusions clear - maybe from the points onwards, because your idea of what you want to convey gets lost in the chemical formulae - and this clouds the picture of your summary.

Basically just that. I have no reservations about this article. Thank you for your diligent work.

Kind regards

Reviewer

Author Response

Modified content attachment sent to you
